# CytA, a reductase in the cytorhodin biosynthesis pathway, inactivates anthracycline drugs in *Streptomyces*

Chun Gui[1,2], Jiang Chen[1,2], Qing Xie[3], Xuhua Mo[4], Shanwen Zhang[1,2], Hua Zhang[3], Junying Ma[1], Qinglian Li[1], Yu-Cheng Gu[5] & Jianhua Ju [1,2]*

Antibiotic-producing microorganism can develop strategies to deal with self-toxicity. Cytorhodins X and Y, cosmomycins A and B, and iremycin, are produced as final products from a marine-derived *Streptomyces* sp. SCSIO 1666. These C-7 reduced metabolites show reduced antimicrobial and comparable cytotoxic activities relative to their C-7 glycosylated counterparts. However, the biosynthetic mechanisms and relevant enzymes that drive C-7 reduction in cytorhodin biosynthesis have not yet been characterized. Here we report the discovery and characterization of a reductase, CytA, that mediates C-7 reduction of this anthracycline scaffold; CytA endows the producer *Streptomyces* sp. SCSIO 1666 with a means of protecting itself from the effects of its anthracycline products. Additionally, we identified cosmomycins C and D as two intermediates involved in cytorhodin biosynthesis and we also broadened the substrate specificity of CytA to clinically used anthracycline drugs.

[1] CAS Key Laboratory of Tropical Marine Bio-resources and Ecology, Guangdong Key Laboratory of Marine Materia Medica, RNAM Center for Marine Microbiology, South China Sea Institute of Oceanology, Chinese Academy of Sciences, Guangzhou 510301, China. [2] University of Chinese Academy of Sciences, Beijing 100049, China. [3] Guangdong Provincial Key Laboratory of Medical Molecular Diagnostics, Institute of Laboratory Medicine, Guangdong Medical University, Dongguan 523808, China. [4] Shangdong Province Key Laboratory of Applied Mycology, School of Life Sciences, Qingdao Agricultural University, Qingdao 266109, China. [5] Syngenta Jealott's Hill International Research Centre, Bracknell, Berkshire RG42 6EY, UK. *email: jju@scsio.ac.cn

Antibiotic producing bacteria that produce toxic agents can develop and employ resistance mechanisms to survive the effects of their products; several major defensive strategies have come to light and these include: efflux pumps, chemical modifications, prodrugs, compound sequestration, subcellular localization, and target modifications/self-resistant protein variants[1]. Anthracycline antibiotics are effective agents in the treatment of various cancers despite their cumulative cardiotoxicity[2]. These compounds consist of a linear tetracyclic polyketide backbone and most are hydroxylated or glycosylated at the C-7 position. Naturally occurring C-7 reduced anthracycline antibiotics are relatively rare (Supplementary Fig. 1). A total of 14 anthracycline antibiotics were purified from marine-derived *Streptomyces* sp. SCSIO 1666[3]. To our surprise, five C-7 deoxyanthracycline antibiotics, cytorhodins X (**1**) and Y (**2**), cosmomycins A (**3**) and B (**4**), and iremycin (**5**)[3–7] (Fig. 1) were identified as major products. Previous reports indicated that deoxyanthracyclines generally show reduced toxicity relative to their C-7 glycosylated counterparts[8]. Compounds **1–10** (Fig. 1) were found to be mycelium-associated whereas fermentation broth supernatants contain less than 5% of the total amount of anthracyclines formed. However, the mechanism that explains the producer's self-resistance to toxic DNA-intercalating agents remains unknown. Our observations here suggest that possible resistance mechanisms likely exist in *Streptomyces* sp. SCSIO 1666. Previous investigations into C-7 reduction have mainly focused on exogenous degradation aspects such as: degradation by mammalian tissue preparations[9], bacterial bioconversion/degradation[10,11], rat liver microsomal NADPH-cytochrome P450 reductase, xanthine oxidase, cytochrome C reductase, and DT-diaphorase-mediated enzymatic reactions in vitro[12]. Microbial conversions of anthracycline antibiotics by bacterial strains (*Aeromonas hydrophila*, *Citrobacter freundii*, and *Escherichia coli*) suggest that these reactions are NADH-dependent and require anaerobic or semi-anaerobic conditions[10,13]. Recently, a pair of enzymes, KstA15 and KstA16 that oxidize the C-1 position of aklavinone from the kosinostatin biosynthetic pathway was identified. Additionally, and somewhat surprisingly, the KstA15/KstA16 pair was also found to catalyze elimination of hydroxyl or glycosyl moieties at the aklavinone C-7 position to afford the shunt product 7-deoxyaklavinone under anaerobic conditions[14].

We have characterized the cytorhodin gene cluster (*cyt*) and elucidated three glycosyltransferases (GTs) involved in tailoring steps from *Streptomyces* sp. SCSIO 1666[3]. As indicated, the *cyt* cluster shows high similarity on the amino acid sequence level (>80% amino acid identity, Supplementary Table 1) to that of cosmomycin D gene cluster (*cos*). Cosmomycin D is decorated with trisaccharide chains at both C-7 and C-10 positions[15] although some key structural differences differentiate the two classes of anthracyclines. The disparity in C-7 oxidation states and substitution patterns between cosmomycin D and cytorhodins X and Y inspired us to hypothesize that an enzymatic process might coordinate formation of the latter. We report here the discovery of CytA, an essential reductase involved in C-7 reduction during the biosynthesis of cytorhodin, a representative anthracycline antibiotic. CytA alone is sufficient to catalyze C-7 reduction under anaerobic conditions and is able to convert clinically employed anthracycline drugs to their inactive, and thereby nontoxic variants. We posit that this unique enzyme and its catalytic chemistry constitute a previously unrecognized resistance mechanism by which *Streptomyces* sp. SCSIO 1666 defends itself from anthracycline toxicities.

## Results

### Bioinformatics analysis reveals enzymatic candidate for C-7 deoxygenation in the cytorhodin biosynthetic gene cluster (BGC).

Comparisons of mammalian and bacterial metabolic systems responsible for anthracycline degradation mechanisms suggest that naturally occurring 7-deoxyanthracyclines **1–5** are generated by specific biosynthetic oxidoreductases. Additionally, heterologous expression studies with the cosmomycin D (bearing a pendant trisaccharide at C-7, Fig. 1) BGC[15], which lacks any discernible CytA homolog, suggested that the enzyme/s driving such chemistry are likely encoded beyond the established *cos* cluster (Fig. 1a); C-7 reduced cosmomycin analogs are not observed among *cos* cluster products. This inspired us to closely examine uncharacterized genes proximal to the *cyt* cluster boundaries. Given that no analogues of the KstA15/KstA16 enzyme pair readily stood out in the *cyt* cluster and that most *cyt* genes have been functionally characterized, we focused our attention on a three-gene cassette containing *cytABC*, the only candidates within the *cyt* cluster that have thus far eluded assignment. The *cytA* gene is located immediately adjacent to the FAD-linked redox encoding gene *cytB* consistent with previously noted organizations (Fig. 1b) for aclacinomycin (*acl*)[16], nivetetracyclate (*niv*)[17], cinerubin (*cin*)[18], and cosmomycin D (*cos*) BGCs[15]. Comparative analyses encompassing the *cyt* as well as *acl*, *niv*, and *cin* clusters reveal that AclJ, NivJ and CinSSBG_00486 are homologues of CytA, and that AknOx, NivO, and CinSSBG_00485 are homologues of CytB. Moreover, CytA (151 aa) shows similarity to AclJ (63% identity), to NivJ (63% identity), and to CinSSBG_00486 (60% identity), respectively. BLAST analyses revealed that CytA, shares amino acid sequence similarities with a family of flavin-dependent reductases (Supplementary Fig. 2) and that CytB is a FAD-linked oxidase whereas CytC is an aldo/keto reductase possibly involved in terminal sugar tailoring. On the basis of these bioinformatics and its genetic organization with respect to the three-gene cassette, we envisioned that CytA may function as a deglycosylase with reductive potential during anthracycline processing. To test this hypothesis, we consequently inactivated *cytA* in vivo and carried out subsequent metabolomics studies.

### In vivo gene inactivation of *cytA* and resultant metabolite profiles analyses.

λ Red-mediated PCR-targeting mutagenesis[19,20] was employed to obtain the Δ*cytA* mutant strain in which *cytA* was replaced with an apramycin resistance gene cassette. The Δ*cytA* mutant was verified on the basis of its sensitivity to kanamycin and apramycin resistance along with PCR analyses (Supplementary Fig. 3). Fermentation of the Δ*cytA* mutant was carried out followed by butanone extractions, solvent removal and resuspension into methanol. The resulting supernatant was then subjected to high-performance liquid chromatography (HPLC) analysis and two new peaks, both having the same UV–Vis profile and representative of **6** and **7** (Fig. 1c) were noted although the overall yield was low. Further liquid chromatography-high-resolution electrospray ionization mass spectrometry (LC-HRESIMS) analyses revealed the molecular formulae of **6** and **7** to be $C_{60}H_{88}N_2O_{21}$ and $C_{60}H_{88}N_2O_{22}$, indicating that **6** and **7** bear one additional trisaccharide chain relative to **3** and **4**, respectively. Large-scale fermentation of the Δ*cytA* mutant strain enabled the isolation of analytically pure **6** and **7** in quantities sufficient for complete structural characterization. $^1H$ and $^{13}C$ NMR spectroscopic data (Supplementary Table 2) for **6** and **7** were found to be identical to previously reported cosmomycins C and D (Fig. 1d)[6], respectively; detailed 2D NMR (COSY, HMQC, and HMBC) analyses further confirmed these structural assignments. Similarly, fermentations with

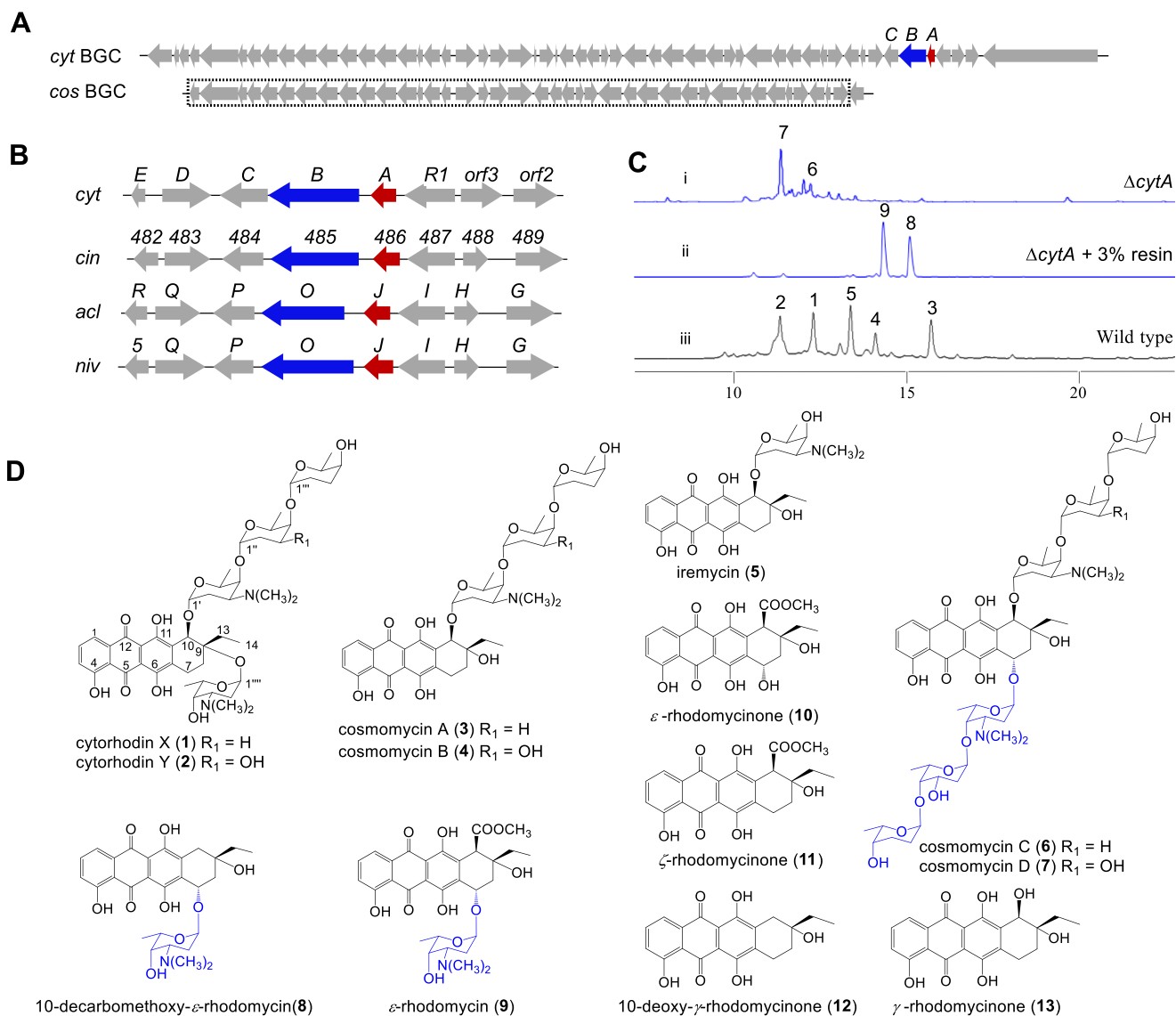

**Fig. 1 Characterization of the deglycosylation step involved in cytorhodin biosynthesis. a** The gene locations of *cytA* in the *cyt* gene cluster. **b** *cytA* and its homologues locations in their respective gene clusters *cyt*, *cin*, *acl*, and *niv* from *Streptomyces* sp. SCSIO 1666, *Streptomyces* sp. SPB74, *Streptomyces galilaeus*, and *Streptomyces niveus* LS2151. **c** HPLC analysis of fermentation broths from (i) mutant strain ΔcytA; (ii) mutant strain ΔcytA supplemented with 3% XAD-16 resin; (iii) the WT strain *Streptomyces* sp. SCSIO 1666/Δ*trdA1*<sub>DH</sub>; **d** Chemical structures of anthracyclines and their aglycones from *Streptomyces* sp. SCSIO 1666. *cytA* and its putative homologs are shaded in red; The captured partial *cos* BGC is shaded in the dotted box. Arrows highlighted in red represented *cytA* and its homologues; arrows highlighted in blue represented *cytB* and its homologues.

the ΔcytA mutant in the presence of 3% XAD-16 resin to capture putative early precursor metabolites, led to the visualization of two additional peaks **8** and **9** with different HPLC retention times (Fig. 1c). Large-scale fermentations employing XAD-16 resin and subsequent NMR studies enabled us to elucidate structures **8** and **9** as 10-decarbomethoxy-ε-rhodomycin and ε-rhodomycin (Fig. 1d, Supplementary Table 3)[21], respectively. That **6**–**9** all contained the anticipated C-7 glycoside supports our hypothesis that CytA represents a previously unreported reductive deglycosylase and inspired subsequent in vitro experiments as follows.

**In vitro biochemical characterization of CytA-catalyzed deglycosylation under anaerobic conditions.** To explore the biochemical activity of CytA in vitro, the *cytA* gene was cloned into the NdeI and EcoRI sites of the pET28a (+) vector and the resulting vector was transformed into *E. coli* BL21(DE3) using previously described methods[22]. CytA was overexpressed as an

N-terminally His₆-tagged soluble protein and purified to homogeneity by Ni affinity chromatography (Fig. 2a). HHpred analysis revealed that CytA is a flavin-dependent protein but the purified protein is colorless. The activity of CytA was first investigated using substrates **6**–**9** and ε-rhodomycinone (**10**) under aerobic conditions. Reactions with CytA were conducted at 37 °C in a volume of 100 μL consisting of 50 mM Tris-Cl (pH 8.0), 25 μM substrate, 0.5 μM CytA, and 20 μM NADH. Reactions with boiled CytA served as controls; all reactions were carried out in parallel to ensure reaction consistency. Reactions were terminated by addition of twofold volume of methanol. As expected, CytA failed, under aerobic conditions to modify any of the substrates **6**–**10**. Carrying out the same reactions using rigorously degassed solutions (see "Methods") afforded very different results. The absence of O₂ enabled CytA to readily convert **6** to **3** and **7** to **4** (Fig. 2b) as revealed by HPLC analyses and comparisons with standards (for **3** and **4**) as well as LC-HRESIMS analyses for each

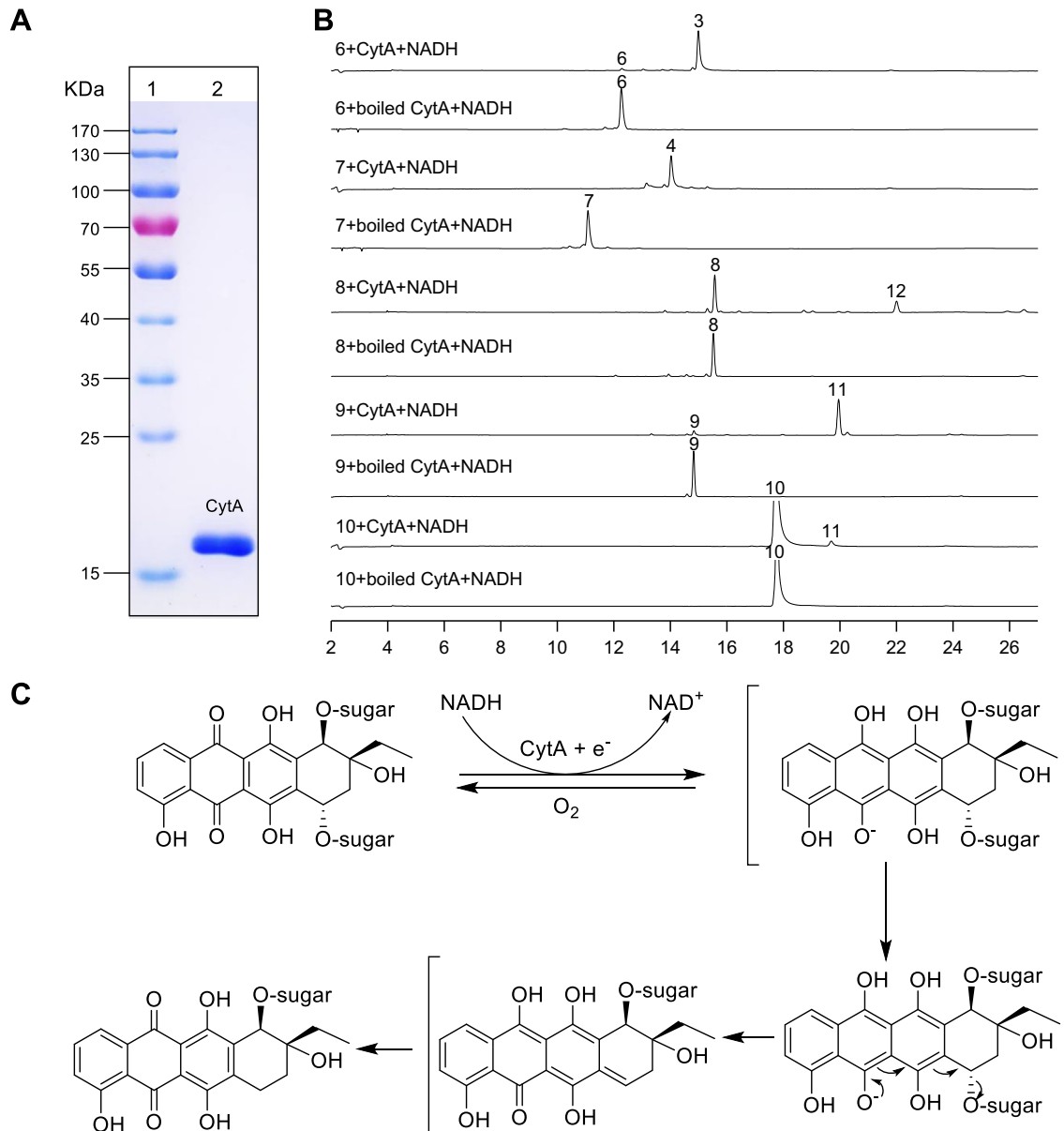

**Fig. 2 Biochemical characterization of CytA in vitro. a** SDS-PAGE analysis of CytA. Lane 1, molecular markers; Lane 2, purified CytA. **b** HPLC analysis of in vitro enzyme assays with CytA and **6**-**10** as reduction substrates. **c** Proposed mechanism for reductive CytA-mediated deglycosylation.

reaction. Under no conditions were deglycosylated or dehydrated intermediates ever observed in CytA reactions. The absence of any such intermediates supports CytA's ability to reductively cleave the C-7 C–O bonds of **6** and **7**; CytA appears to not require hydrolytic or dehydrative chemistries en route to C-7 reduction. On the heels of this finding we then determined the kinetic parameters of CytA using **6** and **7** as substrates. The resulting kinetic data (Table 1) revealed that CytA possesses high catalytic efficiency with $K_m$ of 14.32 and 17.88 μM, and that, of the two substrates evaluated, **6** is preferred. Additionally, CytA showed some enzymatic promiscuity when using **8**–**10** as substrates (Fig. 2b).

**CytA catalyzes reductive deglycosylation of clinically employed anthracycline drugs.** In parallel, a set of C-7-O-glycosylated clinically employed anthracyclines including daunorubicin, doxorubicin, idarubicin, epirubicin and pirarubicin (**22**–**26**) (Supplementary Fig. 4) were screened for their ability to serve as CytA substrates in vitro. These trials, in particular highlighted the

apparent promiscuity of CytA to effect anthracycline deglycosylation chemistry; all five clinical agents were subject to CytA-mediated C-7 reduction/deglycosylatiion albeit with differing efficiencies (Supplementary Figs. 5–7). Enzyme kinetic data for these assays rendered similar $K_m$ values (28.47–49.33 μM, Table 1) for all the clinical anthracyclines evaluated. Overall, CytA was found to be 2–3 fold more efficient at processing initial test substrates **6** and **7** than it was for processing **22**–**26**. Notwithstanding, it was surprising to realize that CytA is less efficient ($K_m$ value 31.18 μM, $k_{cat}$ 0.14 min$^{-1}$) at generating ζ-rhodomycinone (**11**) from **10** given that C-7 hydroxylated aglycones are more sensitive to reduction in sharply contrast to their glycosidic derivatives by exogenous redox enzymes[12]. Overall, these data clearly demonstrate the ability of CytA to serve as a reductive deglycosidase.

**Antibacterial and cytotoxic activity assessments for metabolites 1–13.** We carefully evaluated the antibacterial activities of compounds **1–13** (Fig. 1) using a set of five gram-positive bacteria

**Table 1 Enzyme kinetics of CytA.**

| Substrate | $K_m$ (μM) | $k_{cat}$ (min$^{-1}$) | $k_{cat}/K_m$ (M$^{-1}$ S$^{-1}$) |
|---|---|---|---|
| 6 | 14.32 | 0.71 | 0.82 |
| 7 | 17.88 | 0.79 | 0.73 |
| 8 | 12.70 | 0.17 | 0.23 |
| 9 | 8.50 | 0.33 | 0.65 |
| 10 | 31.18 | 0.14 | 0.08 |
| 22 | 36.29 | 0.58 | 0.27 |
| 23 | 31.01 | 0.61 | 0.33 |
| 24 | 28.47 | 0.44 | 0.26 |
| 25 | 38.57 | 0.53 | 0.23 |
| 26 | 49.33 | 0.74 | 0.25 |

**Table 2 Antimicrobial activities of compounds 1–13.**

| Compound | MIC values (μg/mL) | | | | |
|---|---|---|---|---|---|
| | BS | SA1 | SA2 | BT | EF |
| 1 | 2 | 2 | 2 | 4 | 2 |
| 2 | 1 | 2 | 1 | 4 | 4 |
| 3 | 0.5 | 0.25 | 0.125 | 0.25 | 0.06 |
| 4 | 0.25 | 0.25 | 0.125 | 0.25 | 0.03 |
| 5 | 8 | 8 | 8 | 16 | 8 |
| 6 | 0.06 | 0.03 | 0.005 | 0.125 | 0.001 |
| 7 | 0.03 | 0.01 | 0.005 | 0.125 | 0.002 |
| 8 | 8 | 4 | 8 | 8 | 4 |
| 9 | 8 | 4 | 4 | 4 | 8 |
| 10 | >32 | >32 | >32 | >32 | >32 |
| 11 | >32 | >32 | >32 | >32 | >32 |
| 12 | >32 | >32 | >32 | >32 | >32 |
| 13 | >32 | >32 | >32 | >32 | >32 |
| Van | 0.5 | 0.5 | 1 | 0.5 | 0.25 |
| Amp | 4 | 4 | 8 | 4 | 2 |

*BS B. subtilis BS01, SA1 Staphylococcus aureus ATCC 29213, SA2 Staphylococcus aureus 745524 BT B. thuringiensis BT01, EF E. faecalis ATCC 29212, Van Vancomycin, Amp Ampicillin, Van and Amp served as positive controls. The tests were performed in triplicate*

(Table 2) and found that **6** and **7** were more potent antimicrobial agents than their C-7 deglycosylated counterparts **3** and **4**. Notably, the removal of the sugar moieties at both C-7 and C-9 positions dramatically reduced antibacterial activities (compounds **10** and **11**, MIC values > 32 μg/mL). These findings are consistent with earlier reports in which **6** and **7** were found to be >100-fold more active against *Staphylococcus aureus* than their reduced congeners **3** and **4**, respectively[6]. This seemed at first to also be the trend for cytotoxicity; cosmomycin C (**6**) displayed an IC$_{50}$ of 0.21 μM against MDA-MB-468 cells whereas its C-7 reduced analog cosmomycin B (**4**) was devoid of activity[23]. However, in assessing the cytotoxic activities of **1**–**10** against a panel of tumor cell lines (Table 3) we found that compounds **6** and **7** showed no significant differences in IC$_{50}$ values relative to their deglycosylated counterparts **3** and **4**. At the same time, we noted that cytorhodins X and Y (final biosynthetic products **1** and **2**) bearing the C-9α-glycoside, (absent in both **3** and **4**), were less active in both antimicrobial and cytotoxicity assays than their accumulated intermediates **3**, **4**, **6**, and **7**. This observation strongly suggests that C-7 deglycosylation markedly limits anthracycline bioactivities and that this reaction, especially in light of our findings with CytA, likely constitutes an important self-preservation mechanism for microbial producers of these secondary metabolites.

**CytA confers cosmomycins C and D resistance to the producing organism**. Our in vitro experiments unequivocally show that CytA catalyzes reductive deglycosylation in cytorhodin biosynthesis. We also noted that the Δ*cytA* mutant strain accumulated small amounts (80~100-fold lower than WT producer) of the immediate biosynthetic precursors of cosmomycins A (**3**) and B (**4**). The in vitro antimicrobial activities of compounds **1**–**13** against a panel of gram-positive bacteria (Table 2) provided some insight into the possible physiological significance of CytA-catalyzed reductions. Furthermore, we had observed that the Δ*cytA* mutant strain is clearly impaired relative to the WT strain upon addition of **6** or **7** as potential substrates. To evaluate this effect, we probed the sensitivity of *Streptomyces* sp. SCSIO 1666 to **6** and **7** in an inhibition zone assay (Fig. 3). A clear inhibition zone (16 mm) using the Δ*cytA* mutant plate was observed when supplemented with 1 μg of **6** or **7**. Alternatively, the WT producer appeared to resist the antimicrobial effects of **6** and **7** and displayed little to no growth inhibition near paper discs impregnated with **6** or **7**. That CytA production endows cosmomycin resistance provides strong evidence that the *cytA* gene directly confers *Streptomyces*. sp. SCSIO 1666 with the ability to protect itself from the effects by self-resistance to cosmomycins C and D.

**Phylogenetic analysis of CytA and its homologs**. CytA belongs to a family of reductases widely distributed in Bacteria, Archaea and Eukaryota. Phylogenetic analysis of CytA revealed its

association with incompletely characterized gene products (Supplementary Fig. 8). Notably, most homologous proteins derived from bacteria, especially *Streptomyces* and *Micromonospora*, remain largely uncharacterized. CytA also showed similarity with Eukaryota-derived glycoside hydrolase family 74 protein (40.0% identity) from *Colletotrichum tofieldiae*. The secondary metabolites of these CytA homologues containing microorganisms were largely unexplored. Our work here suggests that CytA and its homologs likely constitute an unidentified self-resistance mechanism associated with anthracycline biosynthetic machineries. As such, these enzymes and their encoding genes may serve as genetic markers for use in genome mining initiatives focused on identifying new anthracyclines from assorted microbial producers (Supplementary Fig. 9).

**Discussion**
The ability of cytorhodin producers to develop self-defense mechanisms to protect them from the effects of self-made cytotoxins such as glycosides **6** and **7** is clear. Under anaerobic conditions, the CytA-catalyzed reaction with NADH first provides a hydroquinone species which then undergoes C-7 deglycosylation to generate a quinone methide; the quinone methide is then envisioned to quickly rearrange to reduction product 7-deoxyanthracycline (Fig. 2c), in a fashion similar to that observed with previously reported transformation reactions[11,24]. In the presence of O$_2$, the initially added electron is transferred from the reduced anthracycline to O$_2$ to return the starting quinone; elimination chemistry at the C-7 center is averted (Fig. 2c).

Microbial transformations and/or exogenous redox enzymes identified in previous studies revealed that several anthracycline antibiotics can be converted to their non-toxic 7-deoxyanthracycline congeners (Supplementary Fig. 4). In this study, we discovered an uncharacterized enzyme that drives the deoxy reduction process. Inactivation of *cytA* in *Streptomyces* sp. SCSIO 1666 impaired the production of C-7 reduced products. Additionally, in vitro studies with purified CtyA revealed its ability to cleave cosmomycins C and D and a set of clinically used drugs with C-7 sugar moieties. Therefore, we conclude that CytA is responsible for degrading cosmomycins C and D to their less active products cosmomycins A and B in the cytorhodin biosynthetic pathway (Fig. 4). The CytA-catalyzed reaction involves anaerobic one electron reduction of the anthracycline quinone

**Table 3 Cytotoxic activities of compounds 1–10 (IC$_{50}$, µM).**

| Compound | A549 | HeLa | HepG2 | RKO | MCF-7 | MDA-MB-231 | MDA-MB-468 | L02 | Huvec-12 |
|---|---|---|---|---|---|---|---|---|---|
| 1 | 3.40 | 2.04 | 3.04 | 11.23 | 12.65 | 18.70 | 21.32 | 3.50 | 4.23 |
| 2 | 7.25 | 3.25 | 6.17 | 11.77 | 9.89 | 17.51 | 16.45 | 4.46 | 4.71 |
| 3 | 0.032 | 0.15 | 0.023 | 0.083 | 0.82 | 1.36 | 0.81 | 0.42 | 0.31 |
| 4 | 0.021 | 0.017 | 0.038 | 0.089 | 0.65 | 0.85 | 0.71 | 0.32 | 0.41 |
| 5 | 11.79 | 9.99 | 10.17 | 15.16 | 10.67 | 20.34 | 29.33 | 6.93 | 13.53 |
| 6 | 0.035 | 0.017 | 0.024 | 0.017 | 0.27 | 0.64 | 0.50 | 0.10 | 0.29 |
| 7 | 0.019 | 0.110 | 0.031 | 0.099 | 0.51 | 0.66 | 1.84 | 0.13 | 0.30 |
| 8 | 6.24 | 1.31 | 1.99 | 3.28 | 21.06 | 7.65 | 16.8 | 19.81 | 6.92 |
| 9 | 1.92 | 1.01 | 1.91 | 2.88 | 11.69 | 6.63 | 13.45 | 13.98 | 7.21 |
| 10 | 33.17 | 20.21 | 17.21 | 18.47 | 37.29 | >50 | 7.43 | 11.79 | 15.31 |
| Dox | 2.31 | 0.83 | 0.41 | 1.45 | 0.90 | 16.85 | 2.11 | 7.67 | 6.31 |
| Cis | 9.31 | 6.07 | 10.27 | 19.96 | 8.93 | 33.11 | 22.23 | 36.73 | 23.02 |

*A549* human lung cancer cell line, *HeLa* human cervical carcinoma cell line, *HepG2* human hepatocellular carcinoma cell line, *RKO* human colorectal carcinoma cell line, *MCF-7* human breast adenocarcinoma cell line, *MDA-MB-231* MDA-MB-468,human triple negative breast adenocarcinoma cell line, *L02* normal human hepatic cell line, *Huvec-12* normal human umbilical vein endothelial cell line, *Dox* Doxorubicin, *Cis* Cisplatin

Wild-type strain          ΔcytA mutant strain

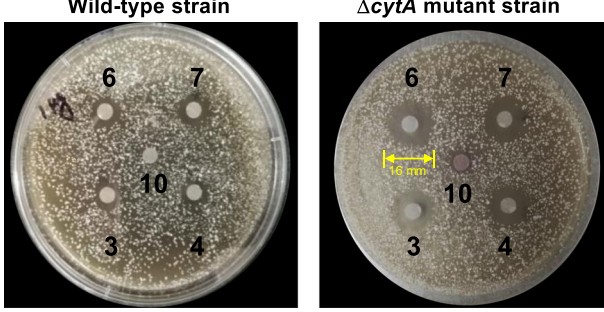

**Fig. 3 Sensitivity of *Streptomyces* sp.** SCSIO 1666 toward cosmomycin C (**6**) and cosmomycin D (7) (1 µg) as determined by growth inhibition zone assays. Compound **10** was inactive against both the WT and Δ*cytA* mutant strains. This growth inhibition zone assays were performed, using varying concentrations of substrates (1–10 ug) and all showed a similar trend of activity.

ring, using NADH as a cofactor; the absence of O$_2$ and longevity of the reduced species enables loss of the trisaccharide sugar moiety. This elimination produces a quinone methide intermediate and subsequent tautomerization by C-7 protonation leads to the formation of the 7-deoxyaglycone. This mechanism of anthracycline detoxification appears to complement previously reported soil-bourne bacterial detoxification systems that rely on respiratory electron transport complex I-like multi-enzyme complexes[11]. We also demonstrated that this enzyme-catalyzed deglycosylation constitutes a mechanism of self-resistance for the antibiotic-producing microbes; that this chemistry has the potential to alter clinically relevant agents was also shown. Hence, to our knowledge, this is the first report of a single enzyme endogenous to the producer that mediates reductive cleavage of anthracycline antibiotics; these studies suggest that CytA plays a key role in both microbial and mammalian mechanisms of anthracycline resistance. Further compounding the importance of these findings is that the CytA homologs AclJ, NivJ, and CinSSBG_00486 are all members of the nitroreductase family which carry out similar chemistries. These enzymes are widespread in Actinobacteria and *Mycobacterium tuberculosis* but are not yet characterized. Thus, beyond its clear relevance to understanding mechanisms of anthracycline resistance, CytA likely provides us a previously unreported tool by which to further our understanding of nitroreductases that have a clear bearing on human medicine.

## Methods

**General experimental procedures**. Bacterial strains and plasmids used in this study are listed in Supplementary Table 4. *E. coli* strains were grown on Lysogeny Broth (LB) medium at 28 °C or 37 °C; appropriate antibiotics were added at a final concentration of: 100 µg/mL ampicillin (Amp), 50 µg/mL apramycin (Apr), 50 µg/mL kanamycin (Kan), 25 µg/mL chloroamphenicol (Cml) and 50 µg/mL trimethoprim (TMP) when necessary. *Streptomyces* sp. SCSIO 1666 was maintained on modified ISP4 medium (M-ISP4, ISP4 medium supplemented with 0.05% yeast extract, 0.1% tryptone and 3% sea salt) plates at 28 °C. Tryptic Soy Broth (TSB) broth was adapted for the spore suspension of *Streptomyces* sp. SCSIO 1666. M11 medium (soybean powder 1%, glycerin 3%, CaCO$_3$ 0.2, 3% crude sea salts, adjusted to pH 7.0) was used for fermentation of *Streptomyces* sp. SCSIO 1666 and mutant strains.

DNA Sequencing was accomplished at Shanghai Majorbio Bio-pharm Technology Co., Ltd (Shanghai, China). Primers were synthesized at Sangon Biotech Company (Shanghai, China). EasyTaq$^{TM}$ DNA Polymerase, TransStart$^{TM}$ Fast-Pfu DNA Polymerase for polymerase chain reactions (PCR) were purchased from TransGen Biotech Company (Beijing, China). Restriction enzymes and DNA ligase were purchased from Takara Biotechnology Co. Ltd. (Dalian, China). Plasmid, gel extraction and cycle-pure kits were acquired from Omega Bioteck Inc. (GA, USA).

Daunorubicin, doxorubicin, idarubicin, epirubicin and pirarubicin were purchased from Aladdin Biochemical Technology Co., Ltd (Shanghai, China). Other chemical solvents (analytical grade) were all purchased from standard commercial sources.

Enzymatic reactions were carried out under anaerobic conditions in an anaerobic workstation (Don Whitley Scientific A45) containing a gas phase of N$_2$/H$_2$/CO$_2$ (80:10: 10%, by vol.). $^1$H, $^{13}$C, and 2D (COSY, HMQC, HMBC and NOESY) NMR spectra were recorded at 25 °C with an Avance 700 MHz spectrometer instruments (Bruker). High-resolution mass spectra were obtained with a Maxis quadrupole-time-of-flight mass spectrometer (Bruker).

**Genomic library screening and annotation of open reading frames**. The genomic library of *Streptomyces* sp. SCSIO 1666 using the SuperCos1 vector has been constructed previously[3,25]. Secondary metabolite BGC were detected and analyzed using online antiSMASH software (http://antismash.secondarymetabolites.org/). The ORFs were determined by application of the FramePlot 4.0 beta program (http://nocardia.nih.go.jp/fp4/). Protein sequences were compared with BLAST programs (http://blast.ncbi.nlm.nih.gov/) and also analyzed by HHpred method (https://toolkit.tuebingen.mpg.de/#/tools/hhpred).

**Inactivation of *cytA* in *Streptomyces* sp. SCSIO 1666 to afford the Δ*cytA* mutant**. The λ Red-mediated PCR-targeting method according to standard procedures[19,26] was performed to inactivate the target gene *cytA*. Primers designed for inactivation are listed in Supplementary Table 5. A general procedure is described as follows. First, cosmid p11D5 which containing the partical *cyt* BGC was transformed into *E. coli* BW25113/pIJ790 for gene inactivations. Then, the gene disruption cassette *aac(3)IV-oriT* was amplified using a fragment from plasmid pIJ773 that was digested with Eco RI and Hind III. PCR products of the *aac(3)IV-oriT* cassette for the disrupted gene were then transformed into *E. coli* BW25113/pIJ790 containing cosmid p11D5 for λ Red-mediated recombination to yield the recombinant cosmid. The recombinant cosmid was transformed into *E. coli* ET12567/pUZ8002 and suffered from conjugation with *Streptomyces* sp. SCSIO 1666 wild-type (WT) strain. Double crossover mutants were selected on the basis

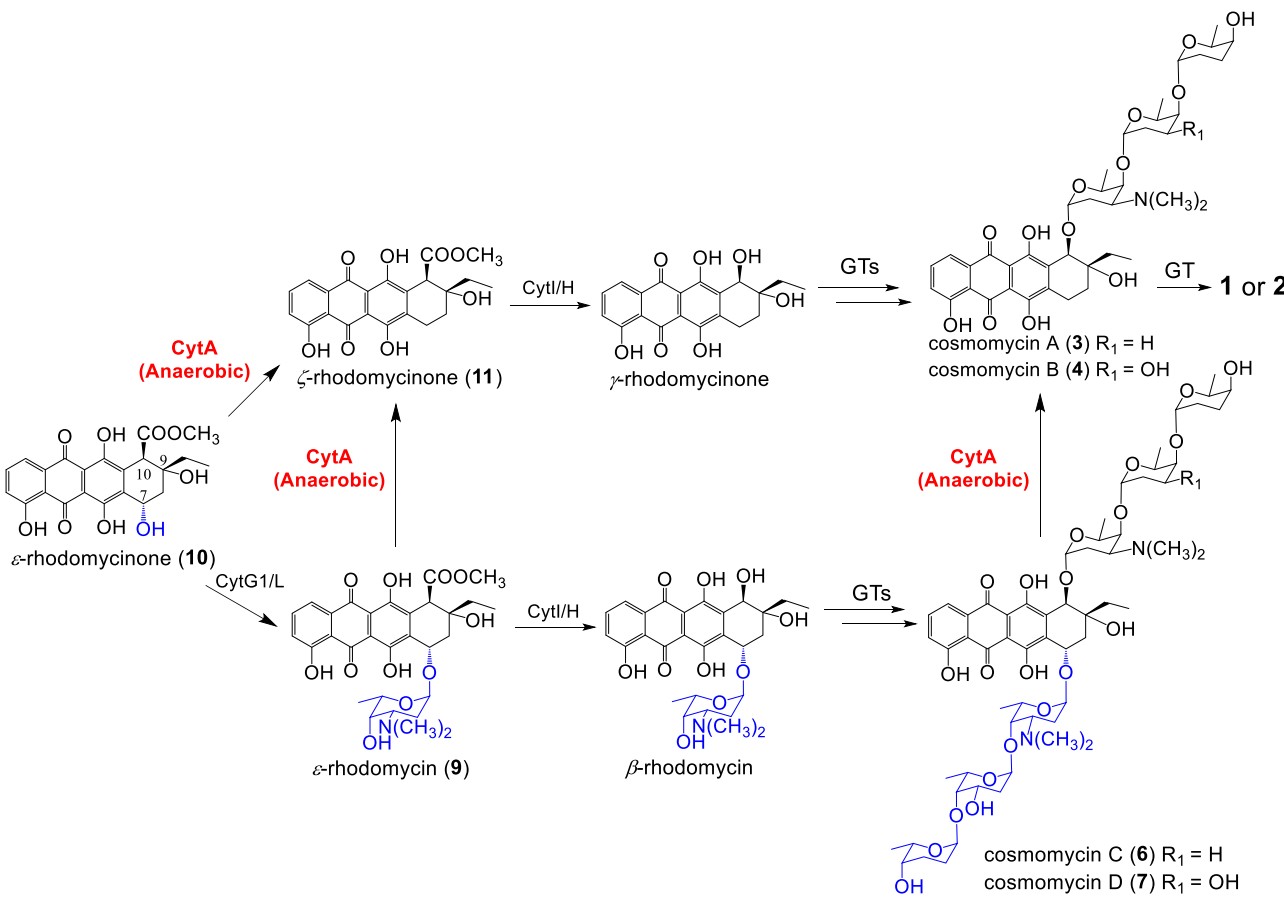

**Fig. 4** Proposed cytorhodin biosynthetic pathway.

of the Kan$^S$Apr$^R$ phenotype and then further confirmed by PCR (Supplementary Fig. 3) using primers listed in Supplementary Table 5. Finally, the $\Delta cytA$ mutant strain of *Streptomyces* sp. SCSIO 1666 was generated.

**Analysis of production profiles of the *Streptomyces* sp. SCSIO 1666 WT and mutant strains**. Unless otherwise stated, the solvent system of analytical HPLC consisted of solvent A (0.1% TFA in ddH$_2$O) and solvent B (100% CH$_3$CN). To analyze the metabolite profiles of *Streptomyces* sp. SCSIO 1666 and its mutants, analytical HPLC was performed on an Agilent 1260 HPLC system (Agilent Technologies Inc., USA) equipped with a binary pump and a diode array detector using a Phenomenex Prodigy ODS column S5 (150 × 4.60 mm, 5 μm) with UV detection at 254 and 500 nm. The samples were eluted with a linear gradient of 5 to 80% solvent B over 20 min, followed by 80 to 100% solvent B in 1 min, and then eluted with 100% solvent B in 5 min, at a flow rate of 1.0 mL/min.

The solvent system used for semi-preparative HPLC consisted of solvent A (50 mM ammonium acetate, pH 8.0) and solvent B (100% CH$_3$CN). Semi-preparative HPLC was accomplished with a Hitachi Model D2000 Elite Chromatography Data Station (Hitachi, Japan) equipped with the Hitachi pump and diode array detector, using an YMC-Pack ODS C8 column (YMC, 250 × 10 mm, 5 μm). Samples were eluted at 2.5 mL/min with a linear gradient from 40 to 90% solvent B for 20 min, 90 to 100% solvent B for 2 min, followed by holding at 100% B for 5 min, and then elution with 40% solvent B for 3 min; UV detection was at wavelengths of 254 nm and 500 nm.

**Large-scale fermentation, isolation and structural elucidation of intermediates 6–9 from $\Delta cytA$ mutant strain**. To characterize intermediates and related compounds from the $\Delta cytA$ mutant strain, a two-step large-scale fermentation strategy was employed. A general procedure is described. First, a suitable portion of spore from solid M-ISP4 medium plate was used to inoculate 50 mL TSB medium in a 250-mL flask as a seed culture; the flask was cultured at 28 °C and 200 rpm for 36-48 h. Then, the seed culture (50 mL) was transferred to 200 mL of fermentation medium (soybean 10 g/L, glycerol 30 g/L, CaCO$_3$ 2 g/L, crude sea salt 30 g/L, adjusted to pH 7.0 ~ 7.2) in a 1000 mL flask; the flask was cultured at 28 °C and 200 rpm for an additional 7–9 days. A total of 30 L culture was carried out. After incubation, the culture broth was harvested and centrifuged to yield a supernatant and a mycelium cake. The supernatant was extracted with an equal

volume of butanone three times and the combined butanone fractions then evaporated to dryness; the mycelium was extracted with 1.5 L acetone three times and organics then evaporated to dryness; the two organic extracts were then combined to yield a residue. The combined residue was then subjected to normal phase silica gel column chromatography, eluted with a gradient elution of CHCl$_3$/MeOH mixture from 100/0, 98/2, 96/4, 94/6, 92/8, 90/10, 80/20, and 50/50 to yield eight fractions (Fr. A1-Fr. A8). The fraction containing the targeted compound on the basis of HPLC analysis was dissolved in MeOH and passed through a 0.22 μm syringe filter followed by semi-preparative HPLC purification to give compounds **6** (3 mg) and **7** (1.8 mg).

For purification of compounds **8** and **9**, the fermentation broth was supplemented with 3% XAD-16 resin during the early stage fermentation. After 7–9 days, the resin was harvested and extracted with 2 L of acetone three times. The solvent was evaporated to dryness to give the crude residue, which was suspended with ddH$_2$O (300 mL), and then extracted with ethyl acetate. The combined organic phase was evaporated to dryness. A similar purification procedure was carried out to finally obtain pure compounds **8** (30 mg) and **9** (28 mg).

**Physicochemical properties and structure Elucidation of compounds 6–9**. Compound **6** had a molecular formula of C$_{60}$H$_{88}$N$_2$O$_{21}$, as determined by HRE-SIMS. The MS, $^1$H and $^{13}$C NMR spectroscopic data of **6** are fully consistent with cosmomycin C in the literature[27]. Thus, the structure of **6** was established.

***Compound 6***: red powder; $^1$H (700 MHz, CDCl$_3$) and $^{13}$C NMR (175 MHz, CDCl$_3$), $^1$H and $^{13}$C NMR data, see Supplementary Table 2; (+)-HRESIMS *m/z* [M + H]$^+$ 1173.5964 (calcd. for C$_{60}$H$_{88}$N$_2$O$_{21}$, 1173.5952); NMR spectra, see Supplementary Figs. 10–15.

Compound **7** had a molecular formula of C$_{60}$H$_{88}$N$_2$O$_{22}$ as indicated by HRESIMS, corresponding to one more hydroxyl group attached to **6**. Analysis of the $^1$H, COSY, HSQC, and HMBC NMR data of **7** allowed full assignment of the NMR signals of **7**. $^1$H and $^{13}$C NMR spectroscopic data of **7** are fully consistent with cosmomycin D data previously reported in the literature[27,28]. Thus, the structure of **7** was established.

***Compound 7***: red powder; $^1$H (700 MHz, CDCl$_3$) and $^{13}$C NMR (175 MHz, CDCl$_3$), $^1$H and $^{13}$C NMR data, see Supplementary Table 2; (+)-HRESIMS *m/z* [M + H]$^+$ 1189.5916 (calcd. for C$_{60}$H$_{88}$N$_2$O$_{22}$, 1189.5901); NMR spectra, see Supplementary Figs. 16–21.

Compound **8** had a molecular formula of $C_{28}H_{33}NO_9$, as determined by HRESIMS, indicating the absence of one -COOCH$_3$ group relative to compound **9**. Analysis of the $^1H$, COSY, HSQC, and HMBC NMR data of **8** (Supplementary Figs. 22–27) allowed full assignment of the NMR signals of **8**. Detailed $^1H$ and DEPT NMR data revealed one more methylene group ($\delta_H$ 3.23, d, J = 19.0 Hz; 2.53, d, J = 19.0 Hz, $\delta_C$ 36.6) was observed in compound **8**. HMBC correlations from H-10 to C-6a, C-8, C-9, and C-10a further confirmed this methylene group at C-10 position. Thus, the structure of **8** was established as a previously unreported compound and named as 10-decarbomethoxy-ε-rhodomycin.

***Compound 8***: red powder; $^1H$ (700 MHz, CDCl$_3$) and $^{13}C$ NMR (175 MHz, CDCl$_3$), $^1H$ and $^{13}C$ NMR data, see Supplementary Table 3; (+)-HRESIMS *m/z* $[M + H]^+$ 528.2249 (calcd. for $C_{28}H_{34}NO_9$, 528.2228), see Supplementary Fig. 6f; NMR spectra, see Supplementary Figs. 22–27.

Compound **9** had a molecular formula of $C_{30}H_{35}NO_{11}$, as determined by HRESIMS. The MS, $^1H$ and $^{13}C$ NMR spectroscopic data of **9** are fully consistent with those of previously reported ε-rhodomycin[29]. Thus, the structure of **9** was established.

***Compound 9***: red powder; $^1H$ (700 MHz, CDCl$_3$) and $^{13}C$ NMR (175 MHz, CDCl$_3$), $^1H$ and $^{13}C$ NMR data, see Supplementary Table 3; (+)-HRESIMS *m/z* $[M +H]^+$ 586.2451 (calcd. for $C_{30}H_{36}NO_{11}$, 586.2283); NMR spectra, see Supplementary Figs. 28–33.

**Overexpression and purification of CytA**. Overexpression of CytA in *E. coli* BL21 (DE3) is described below. The *cytA* gene was PCR-amplified from cosmid p115D with primer pairs of CytA-expF (GGAATTCC**CATATG**GACGAA-CAGGACGAGGTGCT, the underlined sequences represent NdeI site) and CytA-expR (CCGG**GAATTC**TCATCCCCGTGCGGGCCCCT, the underlined sequences represent EcoRI site). The PCR products of *cytA* were recovered from an agarose gel using a gel extraction kit. The corresponding fragments were then excised with NdeI/EcoRI and cloned into the same site of pET/28a(+) vector to yield plasmid pET28a(+)/*cytA*, which were transformed into *E. coli* BL21(DE3) to yield strains *E. coli* BL21(DE3)/pET28a(+)/*cytA* for protein expression.

*E. coli* BL21(DE3)/pET28a(+)/*cytA* was cultured at 37 °C and 200 rpm to OD$_{600}$ = 0.6. Isopropyl β-D-1-thiogalactopyranoside (IPTG) was then added to a final concentration of 0.1 mM to induce the expression of *cytA*. After cultivation at 16 °C for an additional 20 h, the cells were collected by centrifugation, washed with 50 mM Tris-HCl buffer (pH 8.0) twice, resuspended in the binding buffer (50 mM Tris-HCl, 500 mM NaCl, and 5 mM imidazole, pH 8.0), sonicated (0 °C) and centrifuged. The supernatant was loaded onto a 1 mL Ni affinity column packed by Ni-NTA His•Bind Resin, washed with 3 mL washing buffer I (50 mM Tris-HCl, pH 8.0, 500 mM NaCl, 30 mM imidazole, 10% glycerol) and 3 mL washing buffer II (50 mM Tris-HCl buffer, pH 8.0, 500 mM NaCl, 50 mM imidazole, 10% glycerol), eluted by 2.5 mL elution buffer I (50 mM Tris-HCl buffer, pH 8.0, 500 mM NaCl, 250 mM imidazole, 10% glycerol) and 1 mL elution buffer II (50 mM Tris-HCl buffer, pH 8.0, 500 mM NaCl, 1 M imidazole, 10% glycerol). The fractions that were eluted by elution buffer I containing CytA were then desalted with a PD-10 desalting column, concentrated by filtration on a 3 K Amicon Ultra-15 centrifugal filters, and ultimately dissolved in a storage buffer (10% glycerol, 50 mM Tris-HCl buffer, pH 8.0) and stored at −80 °C for further experiments. All protein purification steps were conducted at 4 °C. The concentration of purified CytA was determined using Bio-Rad protein assay dye according to the manufacturer's protocol.

**In vitro biochemical analysis of CytA**. To validate the in vitro biochemical activities of CytA, we overexpressed and purified this enzyme as a soluble N-terminally His$_6$-tagged protein from *E. coli* (Fig. 2a). Tris-Cl buffer was made O$_2$ free by boiling, followed by cooling under N$_2$ flow; NADH stock solution (dissolved in Tris-Cl), substrate stock solution (dissolved in DMSO), and enzyme solution (dissolved in Tris-Cl supplemented with 10% glycerin) were subjected to O$_2$ eliminating process in the anaerobic workstations for at least two cycles (40 min for one cycle). CytA enzymatic activity was tested in 50 μL volume containing 50 mM Tris-Cl (pH 8.0), 25 μM substrate, 0.5 μM CytA, and 20 μM NADH, at 37 °C for 30 min. The reaction was quenched with the addition to 2 volumes MeOH. After centrifugation to remove protein, supernatants were analyzed with Agilent HPLC equipment using a Phenomenex Prodigy ODS column S5 (150 × 4.60 mm, 5 μ) and UV detection at 254 and 500 nm. Samples were eluted with a gradient elution of phase A containing 0.1% TFA in ddH$_2$O and phase B containing 100% acetonitrile, at a flow rate of 1 mL/min.

Kinetic analyses of CytA with putative substrates **6–10**, daunorubicin, doxorubicin, idarubicin, epirubicin and pirarubicin, involved a reaction volume of 20 μL consisting of 50 mM Tris-Cl (pH 8.0), 0.5 μM CytA, 20 μM NADH, and substrate concentrations of 4.5–450 μM.

The kinetics of CytA were measured by determination of the initial velocity for the formation of deglycosylation products with different concentrations of substrates (4.5 μM–450 μM). All assays were performed in triplicate and $K_m$ and $V_{max}$ values were calculated on the basis of curve fitting to the Michaelis-Menten equation: $v = V_{max} *[S]/(Km + [S])$ (Supplementary Fig. 7).

**Antibacterial activities**. The antibacterial activities of compounds **1–13** were assessed using twofold serial dilutions of antibacterial agents in MH broth, according to previously reported standard methods provided by Clinical and Laboratory Standards Institute (CLSI)[30]. These compounds were tested for their antibacterial activities against *Bacillus subtilis* BS01, *Staphyloccocus aureus* ATCC 29213, *Staphyloccocus aureus* 745524, *Bacillus thuringiensis* BT01, and *Enterococcus faecalis* ATCC 29212 using a broth dilution method. Each of the reported MIC values in Table 2 is the lowest concentration of antimicrobial agent that completely inhibits growth of the organism in microdilution wells as detected by unaided eyes.

**Cytotoxic activity assessments for compounds 1–10**. Compounds **1–10** were evaluated for cytotoxic activity using seven human tumor cell lines, A549, HeLa, HepG2, RKO, MCF-7, MDA-MB-231, MDA-MB-468, and two normal cell lines L02, Huvec-12 using MTT method[31]. All data were obtained in triplicate and are presented as means.

**Statistics and reproducibility**. For kinetics of CytA, Origin 9.0 (OriginLab Co.) was used to analyze on the basis of curve fitting to the Michaelis-Menten equation. Experiments were performed in triplicate independently. For antibacterial activity, experiments were performed in triplicate using 96-well plates.

**Reporting summary**. Further information on research design is available in the Nature Research Reporting Summary linked to this article.

## Data availability
Sequence data that support the findings of this study has been deposited in GenBank with accession codes MF773975 for cytorhodin gene cluster. Source data are in Supplementary Data 1. The authors declare that all other relevant data supporting the findings of this study are available within the article and its Supplementary Information files and from the corresponding author upon reasonable request.

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

## Acknowledgements

This study was supported in part by the China NSF (81425022, U1501223, and 41476133), CAS (XDA11030403), Guangdong NSF (2016A030312014), and the Syngenta Ph.D. Fellowship awarded to C. Gui. Additionally, we thank the analytical facility center (Ms. Aijun Sun, Dr. Zhihui Xiao and Mr. Chuanrong Li) of the South China Sea Institute of Oceanology for recording NMR and MS data.

## Author contributions

J.J. and C.G. designed research; C.G., J.C., Q.X., and S.Z. performed the research; C.G., X. M., H.Z., J.M., Q.L., Y.G., and J.J. analyzed data; C.G. and J.J. wrote the manuscript.

## Competing interests

The authors declare no competing interests.
