## [Peer Review File · Communications Biology]

Reviewers' comments:

Reviewer #1 (Remarks to the Author):

The manuscript "An Enzyme Confers Self-Resistance via C-7 Reduction in Anthracycline Antibiotic Biosynthetic Pathway" by Gui et al is an interesting study into the biosynthesis of C-7 deoxygenated anthracycline molecules and proposes a mechanism of detoxification. While the overall work is solid I have some concerns that I believe must be addressed prior to publication. These are listed below.

Major/moderate:

1. The authors assume that CytA is a self-resistance protein. However, an alternate proposal that compounds 8/9 are shunt metabolites from an aborted biosynthesis and cosmomycins A/B and cytorhodin X and Y serve a different function besides antibacterials in the natural environment must be considered and addressed in the manuscript.
2. Line 226-227. The sentence "Phylogenetic analysis of CytA revealed its association with incompletely characterized gene products." Raises a question. Do any of these homologs in the tree occur near "anthracycline-like" BGCs?
3. The PCR confirming the knockout of CytA was performed using oligonucleotides that were part of the KO construct. At least one oligonucleotide must be outside of the homologous recombination region to ensure that integration occurred at the designated site.
4. Figure 4. I cannot see the left panel due to the heliness so it is impossible to interpret the data.
5. Throughout the manuscript (e.g. line 90) the authors say "...C-7 saturated cosmomycin analogs..." This is not technically correct. These analogs would be more appropriately referred to as "deoxygenated at C-7" or "reduced C-7".
6. The paragraph, particularly lines 120-133 is confusing because the authors alternate 6/7 and 8/9. This section could be improved by rewording to help clarify the results obtained. As a side note are 8/9 present in the extract without resin?
7. Line 153. The sentence "Under no conditions were hydrolysis intermediates ever observed in CytA reactions." This sentence is confusing to me. Do the authors mean hydrolysis of the sugar moiety for the C-7 oxygen or introduction of a double bond through an elimination reaction (e.g. E2).
8. For Table S4. I would like to see the literature values included in the table for comparison. The same point should be considered for Table S5.
9. Figure 1 legend. There is no mention of the color scheme used in panel A and B. Please provide a brief description to orient the reader.
10. In Figure 2, panel B. Why are some HPLC traces truncated and missing times before 14 min or after 22 min? Data should not be excluded even if negative.
11. In Figure 2, panel C. Why protonate the phenolic anion generated by reduction? Simply keeping the anion and using this to reform the ketone/perform elimination would be a direct route, no?
12. In Figure 5. The intermediates 9 and 11 are proposed to be transformed to the final

products with GTs according to the figure. However many other transformations are needed including removal of the methyl ester and oxidation of that carbon to provide the alcohol for sugar appendage. Please adjust the figure to reflect this.

Minor:

1. Line 69-70. This sentence "...the cyt clusters show high sequence similarity (> 80% amino acid identity, Supplementary Table 3) to that of cosmomycin D" could be clarified a little through some rewording. I assume the authors mean to compare the gene clusters but it sounds like they are comparing the gene cluster with a different natural product.
2. Line 139 and line 391. Restriction endonuclease names are no longer italicized. See *Nucleic Acids Research*, Volume 31, Issue 7, 1 April 2003, Pages 1805– 1812, <https://doi.org/10.1093/nar/gkg274>.
3. Line 94 and 107. Spell out the number 3.
4. Line 103. "Blast" should be "BLAST"
5. Line 146. Delete "were"
6. Line 267. "Luria-Bertani (LB) medium" is not correct. LB is the abbreviation for Lysogeny Broth.
7. Line 293. "orfs" should be "ORFs"
8. Line 310. " Δ CytA" should be " Δ cytA"
9. Line 316. The model number and components of the Agilent HPLC system are typically listed.
10. The concentration of CytA should be verified by A280 given that determinations by the Bradford method can be significantly different.

Reviewer #2 (Remarks to the Author):

This manuscript mainly reveals that a flavin-dependent reductase CytA-catalyzed reduction can trigger deglycosylation to form a series of C-7 saturated anthracycline antibiotics. They performed *in vivo* and *in vitro* experiments to confirm this. They further showed that these C-7 saturated anthracycline antibiotics have dramatically reduced antibacterial activities, but exhibited no significant difference in cytotoxicity to tumor cell lines when compared to their counterparts with sugars. They also found CytA confers cosmomycin C and D resistance to the producing organism. All these are interesting findings, therefore, I support its publication for the revision after considering the following questions.

1. The title easily misleads readers, because readers are prone to the understanding of direct reduction of the C-7 position by the enzyme. The referee suggests the authors change the title, such as "reductive deglycosylation?"

Moreover, because CytA is a flavin-dependent reductase, the referee suggests that a flavin-dependent reductase or something like this should occur in the Abstract and Introduction.

2. In the abstract, the "Streptomyces" in "S. sp. SCSIO 1666" should not be written as abbreviation. "These C-7 reduced metabolites show reduced antimicrobial and cytotoxic activities relative to their C-7 glycosylated counterparts." is not consistent with the results, because the cytotoxicity to tumor cell lines is not reduced.

3. In the abstract, "identify cosmomycins C and D as two precursors involved in cytorhodin biosynthesis", if CytA is a true resistance enzyme, cosmomycins A and B are actual detoxification products, while cosmomycins C and D should be desired final products but not precursors.

4. "Our observations here suggest that additional resistance mechanisms likely exist in S. sp.

- SCSIO 1666.", the authors used the word "additional", therefore, are there other resistance mechanisms in this producer? If there are, what are they?
5. For "Comparisons of mammalian and bacterial metabolic systems responsible for anthracycline degradation mechanisms suggest that naturally occurring 7-deoxyanthracyclines 1-5 are generated by specific biosynthetic oxidoreductases.", the authors did not give supporting reasons.
 6. The authors may want to change "3 gene cassette" to "three-gene cassette".
 7. Is there some experimental evidence to support that 8/9 is precursor of 6/7?
 8. The authors may want to change "an N-terminus His6-tagged" to "an N-terminally His6-tagged"; "E. coli BL21 (DE3)" to "E. coli BL21(DE3)", no space between BL21 and (DE3), but "NdeI and EcoRI" to "Nde I and Eco RI" with a space in them.
 9. In fig.2, 8, 9, or 10 could also be recognized by CytA, while the main text did not mention that, why?
 10. The authors may want to put fig. 3 into supporting information.
 11. The referee suggests the section "Proposed mechanism of anaerobic deglycosylation" should be put into the Discussion. Otherwise, please do some experiments to support it.
 12. For "As such, these enzymes and their encoding genes may serve as genetic markers for use in genome mining initiatives focused on identifying new anthracyclines from assorted microbial producers.", are there any potential BGC encoding type II PKSs for anthracycline-like polyketides? If there are, the referee suggests that some examples be listed into the Supporting Information.
 13. For "Microbial transformations and/or exogenous redox enzymes identified in previous studies revealed that several anthracycline antibiotics can be converted to their non-toxic 7-deoxyanthracycline congeners", the referee suggests that these 7-deoxyanthracycline congeners be listed in the supporting information.
 14. For "HHpred analysis revealed that CytA is a flavin-dependent protein but the purified protein is colorless.", bioinformatics analysis of CytA revealed it is in fact an F420H(2)-dependent quinone reductase, but the authors performed reactions only using NADH as cofactor, so this is interesting. What are the possible reasons?

Reviewers' comments:

Reviewer #1 (Remarks to the Author):

The manuscript “An Enzyme Confers Self-Resistance via C-7 Reduction in Anthracycline Antibiotic Biosynthetic Pathway” by Gui et al is an interesting study into the biosynthesis of C-7 deoxygenated anthracycline molecules and proposes a mechanism of detoxification. While the overall work is solid I have some concerns that I believe must be addressed prior to publication. These are listed below.

Major/moderate:

1. The authors assume that CytA is a self-resistance protein. However, an alternate proposal that compounds 8/9 are shunt metabolites from an aborted biosynthesis and cosmomycins A/B and cytorhodin X and Y serve a different function besides antibacterials in the natural environment must be considered and address in the manuscript.

Response: We thank reviewer #1 for raising this possibility. We have previously demonstrated that the supplementation of XAD-16 was a useful strategy to capture the possible biosynthetic precursors (*Org. Lett.* 2011, 13, 9, 2212-2215; *J. Am. Chem. Soc.* 2005, 127, 6, 1622-1623). Compounds 8 and 9 were only captured by XAD-16 resin during the fermentation. On the other hand, compounds 8/9 are reasonable intermediates on the basis of most anthracycline biosynthetic pathways (*Top. Curr. Chem.* 2008, 282, 3-74.). Similar to the daunorubicin and aclacinomycin biosynthetic pathway, compound 9 was proposed to be a key intermediate which undergoes decarboxylation to form 8 and subsequent hydroxylation to form β -rhodomycin which was proved to be an on-pathway intermediate during cosmomycin D biosynthesis (*Appl Microbiol Biotechnol.* 2006, 73, 1, 122-131). In our case, enzymes CytI/H are candidates to perform a similar function as in cytorhodin biosynthesis (*Org. Lett.* 2017, 5617-5620) (see Figure 4).

In this manuscript, we demonstrated that compounds 6/7 (cosmomycins C/D) are substrates of CytA and that they show much more potent antibacterial activity compared to cosmomycins A/B and cytorhodin X and Y. Also, sensitivity assays of *Streptomyces* sp. SCSIO 1666 with 6 and 7 showed that the WT producer appeared to be resistant to the antimicrobial effects of 6 and 7 while the Δ cytA mutant was sensitive. Together, these results validated that CytA functions as a self-resistance protein. However, like with many natural products, the biological roles for cosmomycins A/B and cytorhodins X and Y *in vivo* remain unknown, and this should be an interesting project to be studied in the near future.

2. Line 226-227. The sentence “Phylogenetic analysis of CytA revealed its association with incompletely characterized gene products.” Raises a question. Do any of these homologs in the tree occur near “anthracycline-like” BGCs?

Response: We thank reviewer #1 for raising this question. Besides the homologues AclJ/NivJ/CinSSBG_00486 which belong to characterized anthracycline natural product BGCs, there are other candidates in potential “anthracycline-like” BGCs such as WP_018891718.1 from *Streptomyces* sp. CNT-302 (GenBank accession

ARIM00000000.1) and WP_017947349.1 from *Streptomyces* sp. CNS-615 (GenBank accession AQPE00000000.1).

3. The PCR confirming the knockout of *cytA* was performed using oligonucleotides that were part of the KO construct. At least one oligonucleotide must be outside of the homologous recombination region to ensure that integration occurred at the designated site.

Response: We appreciate the reviewer's concern. In this manuscript, we performed the partial knockout for *cytA*. We have double checked the primers used for the mutant confirmation, and we are sure both the primers are outside the homologous recombination region.

4. Figure 4. I cannot see the left panel due to the lines so it is impossible to interpret the data.

Response: We have replaced the original image with a higher-quality image.

5. Throughout the manuscript (e.g. line90) the authors say "...C-7 saturated cosmomycin analogs..." This is not technically correct. These analogs would be more appropriately referred to as "deoxygenated at C-7" or "reduced C-7".

Response: Thank you for the useful suggestion. We have revised the "...C-7 saturated cosmomycin analogs..." as "reduced C-7" throughout the whole manuscript as suggested.

6. The paragraph, particularly lines 120-133 is confusing because the authors alternate 6/7 and 8/9. This section could be improved by rewording to help clarify the results obtained. As a side note are 8/9 present in the extract without resin?

Response: For sake of the clarity, we have re-wording the related sentence to read **"...Similarly, fermentations with the Δ *cytA* mutant in the presence of 3% XAD-16 resin to capture putative early precursor metabolites led to the visualization of two additional peaks 8 and 9 with different HPLC retention times (Fig. 1C). Large-scale fermentations employing XAD-16 and subsequent NMR studies enabled us to elucidate structures 8 and 9 as 10-decarbomethoxy- ϵ -rhodomycin and ϵ -rhodomycin (Fig. 1D, Supplementary Table 4)21, respectively..."**. Additionally, compounds 8 and 9 were not observed in the extract without resin which further supported that they are true precursors of compounds 6 and 7.

7. Line153. The sentence "Under no conditions were hydrolysis intermediates ever observed in CytA reactions." This sentence is confusing to me. Do the authors mean hydrolysis of the sugar moiety for the C-7 oxygen or introduction of a double bond through a elimination reaction (e.g. E2).

Response: We thank reviewer #1 for this suggestion. Accordingly, we have re-phrased the corresponding sentence in the manuscript to read **"Under no conditions were deglycosylated or dehydrated intermediates ever observed in CytA reactions"**.

8. For Table S4. I would like to see the literature values included in the table for comparison. The same point should be considered for Table S5.

Response: We have added all the literature values for comparison.

9. Figure 1 legend. There is no mention of the color scheme used in panel A and B. Please provide a brief description to orient the reader.

Response: We thank reviewer #1 for this suggestion. We have added this annotation in the manuscript as follows. “Arrows highlighted in red represented *cytA* and its homologues; arrows highlighted in green represented *cytB* and its homologues”.

10. In Figure 2, panel B. Why are some HPLC traces truncated and missing times before 14 min or after 22 min? Data should not be excluded even if negative.

Response: We thank reviewer #1 for this suggestion. Accordingly, we have shown the full-length HPLC analysis profiles in the manuscript.

11. In Figure 2, panel C. Why protonate the phenolic anion generated by reduction? Simply keeping the anion and using this to reform the ketone/perform elimination would be a direct route, no?

Response: We thank reviewer #1 for pointing this to us. We have revised the previous one as suggested.

12. In Figure 5. The intermediates 9 and 11 are proposed to be transformed to the final products with GTs according to the figure. However many other transformations are needed including removal of the methyl ester and oxidation of that carbon to provide the alcohol for sugar appendage. Please adjust the figure to reflect this.

Response: We thank reviewer #1 for this suggestion. Full details have been added into the proposed biosynthetic pathway as shown in figure 5 (see in new version figure 4).

Minor:

1. Line 69-70. The sentence “...the *cyt* cluster shows high sequence similarity (> 80% amino acid identity, Supplementary Table 3) to that of cosmomycin D” could be clarified a little through some rewording. I assume the authors mean to compare the gene clusters but it sounds like they are comparing the gene cluster with a different natural product.

Response: We thank reviewer #1 for pointing this out to us. For the sake of clarity, we revised the previous description to “...the *cyt* cluster shows high similarity on the amino acid sequence level (> 80% amino acid identity, Supplementary Table 3) to that of cosmomycin D gene cluster...”.

2. Line 139 and line 391. Restriction endonuclease names are no longer italicized. See *Nucleic Acids Research*, Volume 31, Issue 7, 1 April 2003, Pages 1805–1812, <https://doi.org/10.1093/nar/gkg274>.

Response: We thank reviewer #1 for this suggestion. We have revised all endonuclease names to the standard format.

3. Line 94 and 107. Spell out the number 3.

Response: We thank reviewer #1 for this suggestion. We have revised 3 to three.

4. Line 103. “Blast” should be “BLAST”

Response: We thank reviewer #1 for this suggestion. We have corrected Blast to BLAST.

5. Line 146. Delete “were”

Response: We thank reviewer #1 for this suggestion. We have deleted “were” in Line 146.

6. Line 267. “Luria-Bertani (LB)medium” is not correct. LB is the abbreviation for Lysogeny Broth.

Response: We thank reviewer #1 for pointing this out to us. We have corrected “Luria-Bertani (LB)medium” to “Lysogeny Broth (LB) medium”.

7. Line 293. “orfs” should be “ORFs”

Response: We thank reviewer #1 for pointing this out to us. We have corrected “orfs” to “ORFs”.

8. Line 310 “ Δ CytA” should be “ Δ cytA”

Response: We thank reviewer #1 for this correction. We have corrected “ Δ CytA” to “ Δ cytA”.

9. Line 316. The model number and components of the Agilent HPLC system are typically listed.

Response: We thank reviewer #1 for this suggestion. Accordingly, we have added the details as follows: To analyze the metabolite profiles of *Streptomyces* sp. SCSIO 1666 and its mutants, **analytical HPLC was performed on an Agilent 1260 HPLC system (Agilent Technologies Inc., USA) equipped with a binary pump and a diode array detector** using a Phenomenex Prodigy ODS column S5 (150 × 4.60 mm, 5 μm) with UV detection at 254 and 500 nm.

10. The concentration of CytA should be verified by A280 given that determinations by the Bradford method can be significantly different.

Response: We thank reviewer #1 for this suggestion. The Bradford protein assay reaction is dependent on the amino acid composition of the measured proteins especially the arginine and lysine groups of protein. In our case, CytA shares a similar arginine and lysine amino acid composition (13.5%) as that of BSA (13.6%). Using this standard curve, we believe that the concentration of CytA is reliable.

Reviewer #2 (Remarks to the Author):

This manuscript mainly reveals that a flavin-dependent reductase CytA-catalyzed reduction can trigger deglycosylation to form a series of C-7 saturated anthracycline antibiotics. They performed in vivo and in vitro experiments to confirm this. They further showed that these C-7 saturated anthracycline antibiotics have dramatically reduced antibacterial activities but exhibited no significant difference in cytotoxicity to tumor cell lines when compared to their counterparts with sugars. They also found CytA confers cosmomycin C and D resistance to the producing organism. All these are interesting

findings; therefore, I support its publication for the revision after considering the following questions.

1. The title easily misleads readers, because readers are prone to the understanding of direct reduction of the C-7 position by the enzyme. The referee suggests the authors change the title, such as “reductive deglycosylation?” Moreover, because CytA is a flavin-dependent reductase, the referee suggests that a flavin-dependent reductase or something like this should occur in the Abstract and Introduction.

Response: We thank reviewer #2 for this suggestion. We have revised the title as “an enzyme confers self-resistance via reductive deglycosylation in anthracycline antibiotic biosynthetic pathway”, and we have also followed the reviewer’s suggestion to describe enzyme CytA as a flavin-dependent reductase as reflected both in the abstract and the introduction.

2. In the abstract, the “Streptomyces” in “S. sp. SCSIO 1666” should not be written as abbreviation. “These C-7 reduced metabolites show reduced antimicrobial and cytotoxic activities relative to their C-7 glycosylated counterparts.” is not consistent with the results, because the cytotoxicity to tumor cell lines is not reduced.

Response: We thank reviewer #2 for this suggestion. We have corrected the abbreviation to the full form. Additionally, we have revised the sentence “These C-7 reduced metabolites show reduced antimicrobial and cytotoxic activities relative to their C-7 glycosylated counterparts.” into “These C-7 reduced metabolites show reduced antimicrobial and **comparable** cytotoxic activities relative to their C-7 glycosylated counterparts”.

3. In the abstract, “identify cosmomycins C and D as two precursors involved in cytorhodin biosynthesis”, if CytA is a true resistance enzyme, cosmomycins A and B are actual detoxification products, while cosmomycins C and D should be desired final products but not precursors.

Response: We thank reviewer #2 for this suggestion. For the sake of the clarity, we corrected the sentence “...identify cosmomycins C and D as two precursors involved in cytorhodin biosynthesis...” to “...identify cosmomycins C and D as two **intermediates** involved in cytorhodin biosynthesis...”

4. “Our observations here suggest that additional resistance mechanisms likely exist in S. sp. SCSIO 1666.”, the authors used the word “additional”, therefore, are there other resistance mechanisms in this producer? If there are, what are they?

Response: We thank reviewer #2 for this suggestion. We deleted “additional” in this sentence.

5. For “Comparisons of mammalian and bacterial metabolic systems responsible for anthracycline degradation mechanisms suggest that naturally occurring 7-deoxyanthracyclines 1–5 are generated by specific biosynthetic oxidoreductases.”, the authors did not give supporting reasons.

Response: We appreciate reviewer #2 for this question. We have introduced the background of the mammalian and bacterial metabolic systems responsible for

anthracycline degradation. Inspired by this, we proposed that there might be a similar specific enzyme involved in this transformation in *Streptomyces*.

6. The authors may want to change “3 gene cassette” to “three-gene cassette”.

Response: We thank reviewer #2 for pointing this out to us. We have changed “3 gene cassette” to “three-gene cassette”.

7. Is there some experimental evidence to support that 8/9 is precursor of 6/7?

Response: We thank reviewer #2 for this suggestion. Actually, we and another group (*Organic letters*. 2017, 5617-5620; *Appl Microbiol Biotechnol*. 2006, 73, 1, 122-131) have demonstrated the ΔcytG2 mutant accumulated the decarboxylation product β -rhodomycin (precursor of compounds 8/9), which is an on-pathway intermediate in cosmomycin C/D biosynthesis. Enzymes CytI/H are candidates to catalyze compound 9 to 8 and subsequent hydroxylation product β -rhodomycin (as reflected in figure 4).

8. The authors may want to change “an N-terminus His6-tagged” to “an N-terminally His6-tagged”; “E. coli BL21 (DE3)” to “E. coli BL21(DE3)”, no space between BL21 and (DE3), but “NdeI and EcoRI” to “Nde I and Eco RI” with a space in them.

Response: We thank reviewer #2 for pointing this out to us. Accordingly, we have revised those formats as suggested.

9. In fig.2, 8, 9, or 10 could also be recognized by CytA, while the main text did not mention that, why?

Response: We thank reviewer 2 for this good question. We have added the following sentence to include other substrates involved in the CytA-catalyzed reaction as “**Additionally, CytA showed some enzymatic promiscuity when using 8–10 as substrates (Fig. 2B).**” in the main text.

10. The authors may want to put fig. 3 into supporting information.

Response: We thank reviewer #2 for this suggestion. We have moved fig. 3 into the supporting information as figure S7, and the original supporting information figure S7 has been changed to S8.

11. The referee suggests the section “Proposed mechanism of anaerobic deglycosylation” should be put into the Discussion. Otherwise, please do some experiments to support it.

Response: We thank reviewer #2 for this suggestion. We have put this part into the Discussion.

12. For “As such, these enzymes and their encoding genes may serve as genetic markers for use in genome mining initiatives focused on identifying new anthracyclines from assorted microbial producers.”, are there any potential BGC encoding type II PKSs for anthracycline-like polyketides? If there are, the referee suggests that some examples be listed into the Supporting Information.

Response: We thank reviewer #2 for this suggestion. Three potential BGCs for anthracycline-like polyketides were selected to put into the supporting information as suggested (see Supplementary Figure 9).

13. For “Microbial transformations and/or exogenous redox enzymes exogenous redox enzymes identified in previous studies revealed that several anthracycline antibiotics can be converted to their non-toxic 7-deoxyanthracycline congeners”, the referee suggests that these 7-deoxyanthracycline congeners be listed in the supporting information.

Response: We thank reviewer #2 for this suggestion. We have added 7-deoxyanthracycline congeners reported into the Supplementary part (see Supplementary Figure 7).

14. For “HHpred analysis revealed that CytA is a flavin-dependent protein but the purified protein is colorless.”, bioinformatics analysis of CytA revealed it is in fact an F₄₂₀H₂-dependent quinone reductase, but the authors performed reactions only using NADH as cofactor, so this is interesting. What are the possible reasons?

Response: We thank reviewer #2 for raising this question. Firstly, bioinformatics analysis showed that these family of enzymes belong to the flavin-dependent proteins while few of them were functionally characterized. Secondly, microbial transformations and exogenous redox enzymes-catalyzed reactions suggest the indispensable role of the cofactor NAD(P)H involving in this process. We successfully performed the enzymatic reaction using NADH as cofactor in our study. At the same time, we cannot exclude the possibility that F₄₂₀H₂ is the native cofactor in the CytA-catalyzed reaction.

We would like to thank you and the referees for critical reading and comments of the manuscript. We hope the revised manuscript has met your expectations.

REVIEWERS' COMMENTS:

Reviewer #2 (Remarks to the Author):

I am satisfied with the authors' response and support the new version to be published.

REVIEWERS' COMMENTS:

Reviewer #2 (Remarks to the Author):

I am satisfied with the authors' response and support the new version to be published.

Response: Thanks for reviewer #2's kind comment.